# Regional Differences in Mitochondrial Capacity in the Finger Flexors of Piano Players

**DOI:** 10.3390/jfmk4020029

**Published:** 2019-05-26

**Authors:** Katie Luquire, Kevin K. McCully

**Affiliations:** Department of Kinesiology, University of Georgia, 330 River Road, Athens, GA 30602, USA

**Keywords:** endurance, near-infrared spectroscopy, skeletal muscle, oxidative capacity

## Abstract

Background: Near-infrared spectroscopy (NIRS) has been used to measure oxidative capacity, but regional differences have not been identified. Piano players are also a novel group of subjects for this lab. Methods: Controls (*n* = 13) and piano players (*n* = 8) were tested in a seated position on the right forearm. A fatigue test was performed for three minutes at 2, 4 and 6 Hz using electrical stimulation, which created an endurance index (EI) as the forearm fatigued. A six-cuff oxidative capacity test was performed using manual exercise to activate the muscle and allow for regional specificity. A rate constant (Rc) was generated from the mitochondrial capacity data. Results: Overall, piano players (Rc = 1.76 ± 0.6) and controls (Rc = 1.17 ± 0.3) have significant differences for the last two fingers (*p* = 0.01). While controls have significant differences between the index (Rc = 1.86 ± 0.5) and last two fingers (Rc = 1.17 ± 0.3) (*p* = 0.001), this difference was not observed in piano players. Overall, piano players (EI = 75.7 ± 12.3) and controls (EI = 73.0 ± 17.3) had no differences in endurance index values (*p* = 0.71). Conclusions: Piano players have significant differences in the mitochondrial capacity of the finger flexors that control the last two fingers compared to controls. The lack of difference between groups in the index fingers and overall endurance test suggests playing the piano produces training adaptations to the finger flexor muscles of the last two digits, which are rarely used by control subjects.

## 1. Introduction

The ability of mitochondria to sustain energy during prolonged exercise is an important aspect of resistance to fatigue [1]. Non-invasive assessments of mitochondrial capacity using near-infrared spectroscopy (NIRS) have been well established [2,3]. NIRS-measured mitochondrial capacity measures the transition rate of oxidative metabolism from exercise to rest, similar to methods using magnetic resonance spectroscopy to measure the rate of phosphocreatine resynthesis after exercise [4,5]. Mitochondrial capacity has been associated with increased muscle endurance [6].

Mitochondrial capacity has been measured in the forearm muscles [7,8]. These studies tested the entire forearm muscle group, even though the forearm consists of a number of different muscles that control the different fingers as well as the wrist [9,10]. Playing the piano produces repetitive muscle activities of the forearm muscles, with evidence of overuse injury due to long practice time and physiological stress [11]. Additionally, forearm blood flow increases during piano playing and playing the piano at an increased tempo causes muscle fatigue due to increased exercise intensity [12]. Piano players at the college level are constantly playing pieces of high tempo which could be a stimulus for endurance-related adaptations of the forearm muscles.

The aim of this study was to measure mitochondrial capacity of the regions of the forearm finger-flexor muscles that activate different fingers. Data will be obtained from both piano players and controls (non-piano players). It was hypothesized that piano players will have higher mitochondrial capacity in all of their finger muscles, as well as higher forearm muscle endurance. Furthermore, we hypothesized that the muscle to the index finger would have higher mitochondrial capacity than those to the last two fingers in both groups.

## 2. Materials and Methods

### 2.1. Study Population

Eight piano players and 13 healthy controls participated in this study. The piano players were recruited from the Hugh Hodgson School of Music at the University of Georgia and practiced an average of 19 ± 11.6 h per week. The control group consisted of males and females of similar age, who did not play the piano or perform any regular exercise/activity with their forearm muscles. All participants had no prior history of nerve or muscle disease, sickle cell anemia, or psychological conditions. All subjects gave written, informed consent and STUDY00006112 was approved on 6 December 2018 by the Institutional Review Board at the University of Georgia.

### 2.2. Experimental Protocol

The testing protocol consisted of a muscle-specific endurance test followed by measurements of mitochondrial capacity in two different locations of the right forearm. Each subject was tested in a single day.

### 2.3. Endurance Test

A muscle-specific endurance test was performed as reported previously [13]. For the muscle endurance test, two electrode pads (2.5 cm × 10 cm) were placed on the proximal and distal regions of the forearm with a wireless accelerometer (WAX9, Axivity, Newcastle UK) attached to the skin at the belly of the forearm muscle. Data collection began with a 10-second baseline measurement. Stimulation of the forearm muscle began with the application of electrical twitch stimulation (Theratouch 4.7, Rich-Mar, Chattanooga, TN, USA) for a nine-minute time period consisting of three stages of increasing stimulation frequencies (2, 4, and 6 Hz). Each stage of stimulation frequency lasted three minutes and was followed by a 10-second resting period before moving to the subsequent stimulation frequency. All stimulation frequencies were set at the same level of current (between 25–40 mAmps). The force of muscle contractions was measured as the acceleration generated from the twitch induced surface oscillations. This data was collected by the accelerometer at an acquisition frequency of 400 Hz via a Bluetooth port.

### 2.4. Near-Infrared Spectroscopy (NIRS)

This study used a continuous wavelength NIRS device (Portamon, Artinis Ltd., Einsteinweg, The Netherlands), that provided signals related to oxygen saturation of hemoglobin/myoglobin [14]. Data was collected at 10 Hz from three separation distances. The furthest separation distance (4.5 cm) was used for all analyses. The protocol consisted of checking muscle activation by comparing muscle metabolism during a 30-second ischemic period between rest and finger flexion exercise. Four mitochondrial capacity measurements were performed [2], two each with each set of fingers. The NIRS probe was moved from a middle position on the forearm to a more medial position before the last two fingers were used in order to obtain adequate signals from the finger-specific regions of the flexor digitorum profundus (FDP) and flexor digitorum superficialis (FDS) [9]. This was done for one of the finger positions, then the NIRS device was moved and the protocol was repeated. The position of the NIRS device, blood pressure cuff, and exercise apparatus is shown in Figure 1. An example of the protocol for one subject is shown in Figure 2.

Changes in NIRS signals of oxyhemoglobin and deoxyhemoglobin during ischemic periods are used to measure the metabolism of oxygen in the muscle. The signals from the NIRS device create an exponential curve of recovery rates that give a time constant (Tc) to determine the oxidative capacity of the muscle [15]. In a mitochondrial capacity test, ischemic conditions create changes in the signal that is picked up by the NIRS device. Ischemia is induced by using a rapid cuff inflation protocol with the cuff placed proximal to the muscle that is evaluated. The cuff is inflated to 50–90 mmHg above systolic blood pressure in order to adequately occlude blood flow.

### 2.5. Mitochondrial Capacity Tests

The mitochondrial capacity test consisted of four rounds of six five-second cuffs, and a final resting cuff. Traditional tests of mitochondrial capacity use electrical stimulation (4 Hz, 25–40 mAmps) to activate the muscle, which does not allow for regional specificity. In this study, manual exercise was used to activate muscles specific to the different fingers. Participants lifted a 1 kg weight at a pace of about 2 Hz in 30-second increments. After the four rounds of cuffs were repeated with the last two fingers, there was a five-minute resting period before the final resting cuff in order to allow the muscle to return to pre-exercise conditions. A sample experimental setup is shown in Figure 1.

### 2.6. Data Analysis

Mitochondrial capacity was indicated by the rate constant (Rc) of the recovery of metabolic rate from exercise to rest in the mito6 tests. Data were analyzed in MatLab using custom-written codes. The program allowed for a measurement of slopes from the occlusion periods, which were fit to a single mono-exponential curve to a steady baseline. An example of a curve fit of the rate constant of oxidative capacity test is shown in Figure 3.

Endurance index data from the twitch stimulation protocol was analyzed in Excel, which allowed for the measurement of fatigue from 2 to 6 Hz. The endurance at 6 Hz was determined to be the endurance index (EI) and was used to compare subjects.

### 2.7. Statistical Analysis

A two-way between- and within-subject analysis of variance (ANOVA) was used to evaluate the difference between piano and control group and the differences between fingers.

Significance was accepted at 0.05 alpha level.

## 3. Results

Data were obtained from eight piano players and 13 healthy controls (average practice 19 h per week). The descriptive statistics of the piano players and controls are shown in Table 1.

### 3.1. Mitochondrial Capacity

Figure 3 shows an example of a six-cuff oxidative capacity test and the curve fit. This analysis used the six ischemic slopes and a resting value that was fit to a mono-exponential curve. The rate constant was determined from the curve fit. Four trials of six-cuff tests were obtained for the regions of the forearm that control the index and the last two fingers for every participant. Removing the rate constant that was the most different from the other three did not influence the overall findings, so all four points were averaged, and one mean rate constant was obtained for each participant in both of the two locations.

The rate constants of the finger flexors controlling the index finger and last two fingers were compared between piano players and controls. Overall, piano players (Rc = 2.0 ± 0.8) and controls (Rc = 1.86 ± 0.5) have the same mitochondrial capacities for the finger flexors of the index finger (*p* = 0.74). However, piano players (Rc = 1.76 ± 0.6 min^−1^) and controls (Rc = 1.17 ± 0.3 min^−1^) have significant differences for the forearm regions controlling the last two fingers (*p* = 0.01). While controls have significant differences between the finger flexors of the index (Rc = 1.86 ± 0.5 min^−1^) and last two fingers (Rc = 1.17 ± 0.3 min^−1^) (*p* = 0.001), this difference is not observed in the forearm regions of the index (Rc = 2.0 ± 0.8 min^−1^) and last two fingers (Rc = 1.76 ± 0.6 min^−1^) (*p* = 0.193) of piano players. A summary graph is shown in Figure 4.

### 3.2. Endurance Test

Endurance data was compared between piano players and controls using the 6 Hz endurance index. Overall, piano players (EI = 75.7 ± 12.3%) and controls (EI = 73.0 ± 17.3%) had no differences in endurance index values (*p* = 0.71).

## 4. Discussion

The primary finding in this study was that piano players have significant differences in the mitochondrial capacity of the finger flexors that activate the last two fingers compared to controls. This result is likely because piano players use their last two fingers more often than controls, as playing the piano requires stretching and force production of the last two fingers. Exercising the finger flexor muscles leads to tension, microtrauma and fatigue in the muscle and produces changes in cardiac output, blood flow, and heart rate [12]. Therefore, playing the piano, especially at vigorous levels, can be seen as a form of endurance exercise for the finger flexor muscles, particularly to the muscles of the last two fingers. The flexor digitorum superficialis (FDS) runs from the elbow, along the forearm, under the wrist, to each finger and the flexor digitorum profundus (FDP) runs along the elbow down to the forearm to the first finger [10]. Playing the piano for as little as three hours per week has been indicated as a stimulus for upper extremity musculoskeletal symptoms [16]. Participants in this study practiced an average of 19 h per week, which has the potential to cause significant adaptations in the muscle. Endurance exercise has been shown to improve mitochondrial efficiency by improving the mitochondrial phosphate/oxygen ratio, mitochondrial biogenesis, and mitochondrial electron transport chain protein content [17]. These muscular adaptations lead to an increased overall mitochondrial capacity after exercise.

A three-dimensional kinetic analysis of hand movement during piano playing shows that the index and middle fingers are most often used in piano playing, but the last two fingers are used 13% of the time in piano students [18]. While this statistic appears meager, piano players in this study recorded an average practice time of 19 h per week. Even if piano students only use their last two fingers 13% of the time, about 2.5 h of practice with their last two fingers is dedicated to exercise of the last two fingers each week. This quantity of practice is likely significant enough to produce the observed mitochondrial differences.

It was determined that there were no differences in the mitochondrial capacity of the finger flexors controlling the index finger between piano players and controls. Additionally, the mitochondrial capacity of the finger flexors controlling the index finger was significantly higher than the muscles controlling the last two fingers in controls. The index finger is regularly used in everyday life, so both piano players and controls receive a high level of training and, therefore, have similarly trained index finger flexors. A study on hand kinematics showed significant use of the index and middle fingers during manipulation of a feasible workplace, which supports the hypothesis that index finger usage in controls contributes to a higher oxidative capacity [19].

Additionally, there were no significant differences in the endurance index between piano players and controls. It is possible that the electrically-induced muscle contractions activated the index portion of the finger flexor muscles more than the last two fingers which would have caused the resulting endurance index to be more similar between the two groups. When comparing piano players and controls, the statistical significance is similar between the index finger flexor mitochondrial capacity (*p* = 0.74) and the endurance index (*p* = 0.71), indicating that the two tests produced analogous results.

One limitation of the study was a limited sample of piano players. Participants in this study ranged from 10–40 h of practice per week, but there were only eight piano students who participated in this study. We did not collect information on whether the piano players used other instruments or performed other training in addition to playing the piano. When piano player mitochondrial capacity was compared to the number of hours practiced per week, no discernible pattern was seen for either the finger flexors controlling the index or middle fingers. Future studies could explore how the amount of practice per week influences finger flexor mitochondrial capacity, but more participants of varying practice levels would need to be recruited.

## 5. Conclusions

In conclusion, regional differences in mitochondrial capacity can be measured in the forearm muscles using manual exercise to activate the muscle of interest. Piano players have significant differences in the region of the finger flexors of the last two fingers compared to controls, whereas controls have significantly higher mitochondrial capacities in the finger flexors that control the index finger relative to the last two fingers. This study indicates that there can be regional localization of oxidative capacity and this protocol can be used to compare specific populations.

## Figures and Tables

**Figure 1 jfmk-04-00029-f001:**
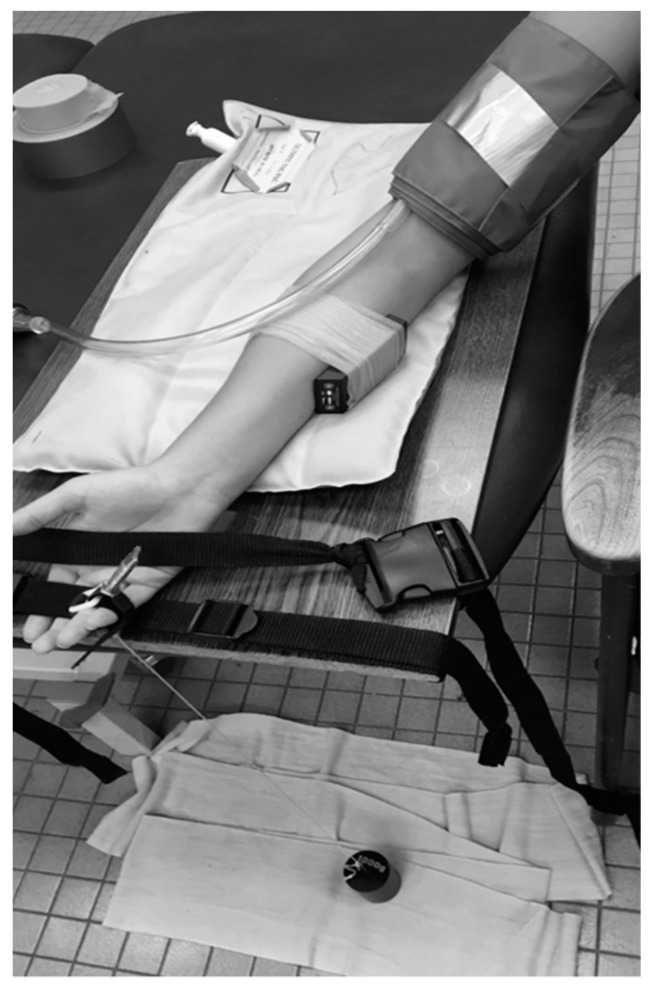
Experimental arrangement. The near-infrared spectroscopy (NIRS) device was placed in the medial (as shown) and lateral positions of the body of the forearm flexor muscles. A blood pressure cuff was placed proximal to the NIRS device. Manual exercise with a 1000 g weight was used to activate finger flexor muscles individually. Subjects were tested in a seated position with a pad under the elbow for support.

**Figure 2 jfmk-04-00029-f002:**
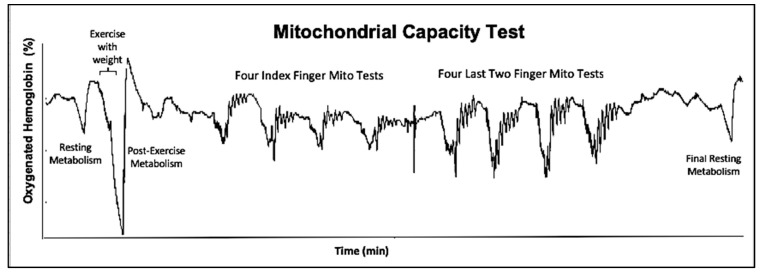
NIRS output produced by the full mitochondrial capacity test. Includes one resting slope, exercise with a weight followed by an exercise cuff, four trials of six cuffs for both the index and last two fingers, and a final resting slope. Slopes decrease over time, indicating that there is recovery of oxygen consumption.

**Figure 3 jfmk-04-00029-f003:**
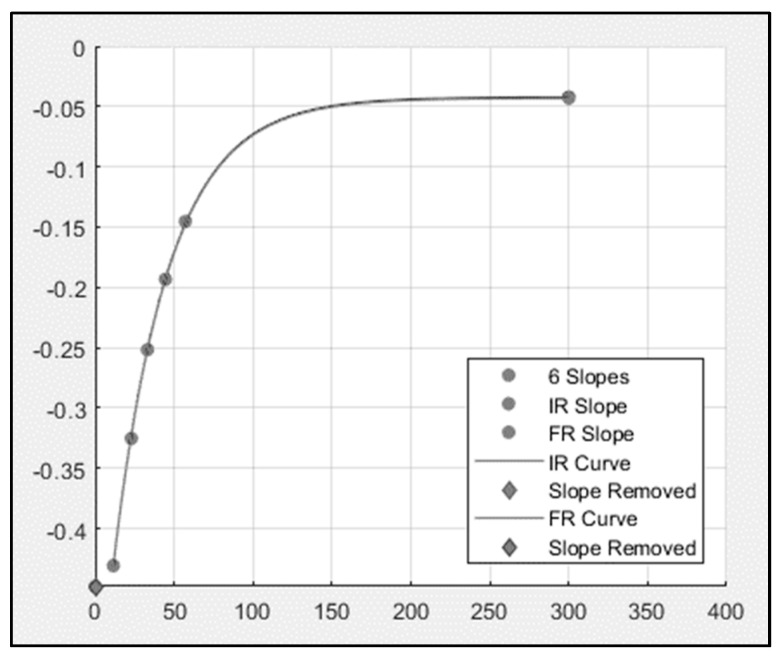
MatLab curve fit. The six slopes from one six-cuff trial were plotted in MatLab and fit to an exponential curve at a steady baseline. The resulting rate constant (Rc) is proportional to mitochondrial capacity. Slopes are measured as mitochondrial rate capacity (1/min) and Rc = 1.61.

**Figure 4 jfmk-04-00029-f004:**
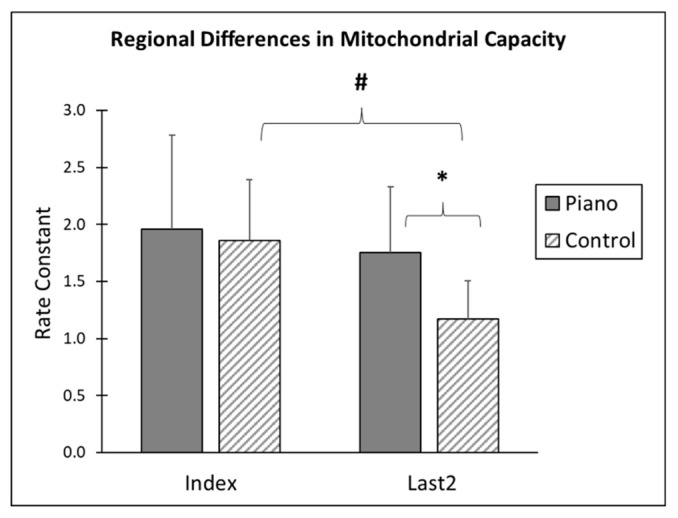
Regional differences in mitochondrial capacity. Control index Rc = 1.86 ± 0.5 and control last two Rc = 1.17 ± 0.3 (*p* = 0.001). The last two fingers in piano players showed Rc = 1.76 ± 0.6 and in controls Rc = 1.17 ± 0.3 (*p* = 0.01). A 2 × 2 analysis of variance (ANOVA) shows a significant difference between piano players and controls (F = 5.0, *p* = 0.039), and a significant difference between index and last two finger flexors (F = 12.8, *p* = 0.02). * and # denote significance.

**Table 1 jfmk-04-00029-t001:** Relevant participant demographics for piano players and controls.

Subject	Gender	Age (yr)	Weight (kg)	Height (cm)	BMI (kg·m^−2^)
Piano players	6F/2M	21.9 ± 4.1	60.5 ± 7.0	169.2 ± 7.6	21.1 ± 2.1
Controls	7F/6M	20.8 ± 0.9	69.4 ± 12.3	172.9 ± 11.1	23.1 ± 2.6

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
