# Peer review of "Regional Differences in Mitochondrial Capacity in the Finger Flexors of Piano Players"

_jfmk, 2019, doi:10.3390/jfmk4020029_

Round 1
Reviewer 1 Report
The manuscript describes mitochondrial rate capacity of fingers and forearm muscle endurance in piano players. The study is very concise yet of interest, but the text needs some revision to make it suitable for publication.
Major remarks
1. Information on lifestyle of subjects needs to be expanded. Were both arms evaluated, or the dominant arm only? Did piano players also exercise another instrument of have a profession that requires intensive typing? The latter question also concerns the control subjects. Please add this information to the study population section. Also, why were piano players chosen? Perhaps it would have been a good idea to work with cello players and compare left and right hand, as within one subject very different movement are then combined. Please comment.
2. In figure 2 , a representative control and piano player profile should be shown.
3. The conclusion section contains a lot of repetition. Please shorten substantially, and also remove redundant lines 209-212 as this concerns ‘conclusion’ in the discussion section.
4. Some unclear entries: What do the authors mean with the sentence ‘Piano players are also a novel group of subjects for this lab’ in the abstract. Nd the sentence ‘This suggests as a group, … muscle.’ In the conclusion section (line238)?
Minor remarks
1. line 58: by mitochondrial
2. line 172: to each finger
3. line 173: piano for as little
4. line 244: et al. have shown
5. reference 6 lacks page information
Author Response
The manuscript describes mitochondrial rate capacity of fingers and forearm muscle endurance in piano players. The study is very concise yet of interest, but the text needs some revision to make it suitable for publication.
We appreciate the efforts of the reviewer in evaluating our manuscript. We have revised our paper and feel it has been improved.
Major remarks
1. Information on lifestyle of subjects needs to be expanded. Were both arms evaluated, or the dominant arm only? Did piano players also exercise another instrument of have a profession that requires intensive typing? The latter question also concerns the control subjects. Please add this information to the study population section. Also, why were piano players chosen? Perhaps it would have been a good idea to work with cello players and compare left and right hand, as within one subject very different movement are then combined. Please comment.
We only evaluated the right arm, as the right hand typically performed more ‘work’ than the left hand in piano. We do not have information on other activities or instruments the piano players may have used. Because our subjects played the piano an average of 19 hours a week, we assumed they were not spending too much additional time playing other instruments. We agree that other musical populations might be of interest to study, we piano because of our ability to recruit subjects, and because playing the piano does require using the finger flexor muscles.
We have revised the methods to clarify the arm tested, and the limitations of the study to include the possibility of other musical activities.
2. In figure 2 , a representative control and piano player profile should be shown.
We have chosen not to add another figure to the paper. We understand the reasons why this might be useful. However, the differences in rate constants seen in our study are relatively small, and the fitted curves (such as figure 2) would look very similar. We show the curve to highline the methodological approach used to obtain the rate constants. Other papers where recovery curves are shown for groups with >100% differences in rate constants have been reported and do look reasonable.
3. The conclusion section contains a lot of repetition. Please shorten substantially, and also remove redundant lines 209-212 as this concerns ‘conclusion’ in the discussion section.
We have followed the reviewer’s recommendations and shortened this section.
4. Some unclear entries: What do the authors mean with the sentence ‘Piano players are also a novel group of subjects for this lab’ in the abstract. Nd the sentence ‘This suggests as a group, … muscle.’ In the conclusion section (line238)?
We have revised this section as suggested. The conclusion section did repeat what had been said earlier and has been taken out of the revised manuscript.
Minor remarks
These comments have been addressed
1. line 58: by mitochondrial
2. line 172: to each finger
3. line 173: piano for as little
4. line 244: et al. have shown
5. reference 6 lacks page information
Reviewer 2 Report
Overall, this is a really interesting study showing some differences between piano players and non-players. Line 78 I'm not sure what "mito6" is. Line 81 you have undefined abbreviations. they are defined later in the discussion, but they should be defined here. Results: I don't see units for the rate constant or the endurance index.
Author Response
We appreciate the efforts of the reviewer to evaluate our manscript
Line 78, we have revised and provided a reference
Line 81, we have defined these abbreviations
Results now include units for rate constants.
Round 2
Reviewer 1 Report
Please specify in the text that all subjects were right handed.